# Allometric Equations to Estimate Aboveground Biomass in Spotted Gum (*Corymbia citriodora* Subspecies *variegata*) Plantations in Queensland

Trinh Huynh [1,2,*], Tom Lewis [1,3], Grahame Applegate [1], Anibal Nahuel A. Pachas [1,3] and David J. Lee [1]

1 Forest Research Institute, University of the Sunshine Coast, Locked Bag 4, Maroochydore, QLD 4558, Australia; tlewis1@usc.edu.au (T.L.); gapples@usc.edu.au (G.A.); nahuel.pachas@daf.qld.gov.au (A.N.A.P.); dlee@usc.edu.au (D.J.L.)
2 Forest Science Institute of Central Highlands and South of Central, Da Lat City 670000, Vietnam
3 Department of Agriculture and Fisheries, Queensland Government, 1 Cartwright Road, Gympie, QLD 4570, Australia
* Correspondence: trinh.huynh@research.usc.edu.au

**Abstract:** Accurate equations are critical for estimating biomass and carbon accumulation for forest carbon projects, bioenergy, and other inventories. Allometric equations can provide a reliable and accurate method for estimating and predicting biomass and carbon sequestration. Cross-validatory assessments are also essential to evaluate the prediction ability of the selected model with satisfactory accuracy. We destructively sampled and weighed 52 sample trees, ranging from 11.8 to 42.0 cm in diameter at breast height from three plantations in Queensland to determine biomass. Weighted nonlinear models were used to explore the influence of different variables using two datasets: the first dataset (52 trees) included diameter at breast height (D), height (H) and wood density ($\rho$); and the second dataset (40 trees) also included crown diameter (CD) and crown volume (CV). Cross validation of independent data showed that using D alone proved to be the best performing model, with the lowest values of AIC = 434.4, bias = −2.2% and MAPE = 7.2%. Adding H and $\rho$ improved the adjusted. $R^2$ ($\Delta$ adj. $R^2$ from 0.099 to 0.135) but did not improve AIC, bias and MAPE. Using the single variable of CV to estimate aboveground biomass (AGB) was better than CD, with smaller AIC and MAPE less than 2.3%. We demonstrated that the allometric equations developed and validated during this study provide reasonable estimates of *Corymbia citriodora* subspecies *variegata* (spotted gum) biomass. This equation could be used to estimate AGB and carbon in similar spotted gum plantations. In the context of global forest AGB estimations and monitoring, the CV variable could allow prediction of aboveground biomass using remote sensing datasets.

**Keywords:** biomass prediction; crown volume; cross-validatory assessment; destructive sampling; hardwood plantation; weighted nonlinear models

## 1. Introduction

The ability to accurately estimate biomass and carbon will impact any incentive program using forests as part of the solution for emissions reductions [1,2]. Uncertainty in biomass and carbon estimates, resulting from a lack of species-specific allometric equations to accurately determine biomass from easily measurable parameters such as tree diameter or height, needs to be addressed [3]. Many scientists reported that maintaining and expanding forests will play a key role in storing carbon and in removing carbon from the atmosphere to assist in securing global net-zero emissions of greenhouse gasses and keep the earth from increasing its average temperature by 1.5 degrees [4]. As part of the commitment or National Determined Contributions (NDC) under the United Nations Framework Convention on Climate Change (UNFCC) framework, each country is required to quantify its forest carbon sources and sinks, which are determined through national inventories of net greenhouse

gas emissions and forest carbon budgets [5]. The Intergovernmental Panel on Climate Change (IPCC) recommended that for determining carbon capture and storage for different forest types, countries should adopt an appropriate level of reporting from one of three tiers. These tiers range from simple (Tier 1) to complicated methods (Tier 3) to calculate the carbon capture and storage [6]. Each country is encouraged to develop Tier 2 and 3 methodologies, particularly for new sources or carbon sinks, and these Tiers are considered important in the 2019 refinement to the 2006 IPCC Guidelines for National Greenhouse Gas Inventory [7].

Establishing aboveground biomass (AGB) allometric equations for spotted gum (*Corymbia citriodora* subspecies *variegata*, CCV) will allow plantation owners to track carbon accumulation more confidently, important for both timber production and trading carbon credits. This Australian native species provides a high-value hardwood timber in New South Wales and Queensland, Australia [8,9], and is known to tolerate a range of climates, making it a desirable species as weather patterns change and become more extreme [10]. Globally, this species is being planted in South Africa, South-East Asia, South and North America, Brazil, and Israel [11,12]. Planted spotted gum forests have the potential to be large carbon sinks and contribute to mitigating climate change. Initial studies in Queensland indicate that spotted gum plantations can store at least 100 t ha$^{-1}$ of $CO_2$ equivalents in the main stem of the trees by age 10 [13]. The amount of $CO_2$ equivalents accumulated was up to 184 t ha$^{-1}$ and 159 t ha$^{-1}$ at an age of 10 years in the coastal Wide Bay Burnett and Central Coast—Whitsunday regions, respectively in Queensland [13]. However, the lack of allometry based on destructive sampling of trees biomass could result in inaccurate estimates of biomass and carbon accumulation for mature plantations and the estimates of Lee et al. (2011) [13] did not account for the carbon in other tree components (such as branches and leaves).

Accurate estimation of tree biomass over time provides various benefits in predictions of $CO_2$ sequestration rates [14], production of biofuels, and electricity using biomass residues [12]. Several allometric models for estimating forest biomass have been established worldwide, and there are over 400 models for biomass estimation in natural forests and plantations in Australia [3,15]. The majority of these estimates were developed for natural forests such as those dominated by *Eucalyptus* species [3,16–21]. Biomass datasets derived through destructive sampling for spotted gum (CCV) are relatively rare and have not been published in Queensland. A study by Garcia-Florez et al. (2019) [12] sampled 16-year-old plantation grown CCV trees southwest of Lismore in north-eastern New South Wales, Australia, to estimate individual above-ground biomass components (stem, branches, bark, and crown) but not total AGB. In a native forest near Batemans Bay in southern coastal New South Wales, Ximenes et al. (2006) [22] destructively sampled 122 spotted gum (*Corymbia maculata*) trees to predict total AGB and commercial log biomass. However, it is difficult to ascertain whether these models could be fit in spotted gum plantations in Queensland.

Biomass estimates are often based on traditional field inventory methods that involve measuring tree attributes such as diameter at breast height, and these can provide accurate estimations where an appropriate number of inventory plots can be sampled [3]. However, AGB can also be estimated on a regional scale using remote sensing data in combination with allometric models [2]. Remote sensing data, such as LiDAR and satellite-based photogrammetry can provide reliable estimates of tree heights, basal area and canopy dimensions [23,24] over large areas with relatively high cost-effectiveness [25] and low levels of uncertainty [26]. If reliable allometrics can be derived using these variables, then tree and hence forest stand biomass could be estimated over large spatial extents, without the need for field sampling [1,27]. In both cases, allometric equation development is the starting point for addressing information gaps and improving biomass estimates.

Developing and selecting appropriate AGB models involves several steps. These include choice of model, selection of independent variables [28,29], and validating the application of selected models with independent data [30–32]. Independent variable choice (e.g., diameter or height) impacts the reliability of the derived allometric equation for

biomass estimations. Chave et al. (2005) [28] found that allometric functions are commonly developed using combinations of two easily measurable parameters; tree size, or diameter (D) and height (H), and these explain most of the variation in AGB estimates [33–37]. Other parameters including wood density ($\rho$) and canopy variables have also been used [14,38,39]. Increasing the number of independent variables from one variable (D) to two variables (D and H) or a combination of three variables (D, H and $\rho$) or four variables (D, H, $\rho$ and canopy variable) results in allometric models with decreasing error and increasing reliability [40]. However, collinearity and model over-parameterization may occur when adding too many variables [41]. Additionally, the high costs associated with measurement of certain variables (e.g., $\rho$ and canopy) may limit the widespread use of allometric equations based on these variables. An alternative approach to maximize the cost-effectiveness is validating new data using training and testing datasets to provide confidence when applying equations at new sites [18,30,31,42]. Hence, an appropriate allometric model should consider collinearity issues [41], time, cost restraints, the objectives of the model (i.e., appropriate tree parameters for the users) [43] and testing against new data.

Our objectives were to determine which tree variables improved the accuracy of biomass estimates using allometric equations and to indicate the most appropriate allometric equation(s) for predicting aboveground biomass in CCV plantations. We aim to answer the following three research questions: (i) What is the influence of different independent variables such as tree height, wood density and crown variables on biomass estimation? (ii) What is the best model for estimating AGB following cross validation? and (iii) What allometric models are best at estimating AGB in spotted gum plantations?

## 2. Materials and Methods

### 2.1. Study Area and Datasets

The study was conducted in three CCV research trials (451D, 451G and 13PHY) in southeast Queensland, Australia. A description of three plantation sites was presented by Huynh et al. (2021) [44].

Datasets of AGB were selected based on three stands of 7, 8, 9, 18, and 20-year-old healthy trees with single stems [45]. The data were collected during two periods. In 2009, the first dataset consisted of 12 trees sampled in 451D, 451G and 13PHY (D ranges 11.8–18.2 cm); and in 2020, the second dataset consisted of 40 sample trees from 451D and 451G (D ranges 17.1–42.0 cm). The number of sample trees and description of these sites is given in Table 1 and Figure S1.

**Table 1.** Study sites, stand age, summary of predictors of sample trees for developing allometric equations. Abbreviations as follows: n, total number of trees sampled; D, diameter at breast height (cm); H, total tree height (m); CD, crown diameter (m); and $\rho$, wood density (kg m$^{-3}$). For each of D, H, CD and $\rho$, mean, minimum and maximum values are indicated.

| Sites (Age) | n | Mean (min, max) | | | |
|---|---|---|---|---|---|
| | | D (cm) | H (m) | CD (m) | $\rho$ (kg m$^{-3}$) |
| 451G (7) | 3 | 17.8 (11.8–17.6) | 17.4 (15.3–20.4) | NA | 702.6 (646.8–752.8) |
| 13PHY (8) | 6 | 15.3 (12.5–18.2) | 15.4 (13.1–16.4) | NA | 676.7 (613.0–738.8) |
| 451D (9) | 3 | 14.4 (12.0–17.8) | 15.5 (12.6–17.5) | NA | 663.8 (631.1–713.1) |
| 451G (18) | 13 | 27.1 (17.6–39.9) | 27.0 (22.1–29.9) | 5.3 (3.0–7.9) | 730.8 (671.5–813.5) |
| 451D (20) | 27 | 28.6 (17.1–42.0) | 25.8 (20.2–32.0) | 6.1 (2.8–9.9) | 736.5 (625.7–801.0) |
| Total | 52 | 25.9 (11.8–42.0) | 23.9 (12.6–32.0) | 5.9 (2.8–9.9) | 722.0 (613.0–813.5) |

### 2.2. Data Collection

In 2009, biomass data were collected in young trees (7–9 years old) at three sites 451D, 451G and 13PHY. In 2020, a similar methodology was conducted in mature trees (18–20 years old) in 451D and 451G as outlined in Huynh et al. (2021) [45] were destructively

sampled to obtain individual tree component weights for AGB biomass estimates. In summary, the procedure for collecting AGB in this study was as follows:

1.  Prior to sampling, each sample tree was identified and provided with an ID number. Tree diameter over bark at breast height 1.3 m (D, cm) was recorded.
2.  Most of trees were felled using a chainsaw. However, an excavator was used to push 23 trees onto the ground as these trees were also used to determine belowground biomass [44]. After the tree was felled, total tree height (H, m) was recorded.
3.  Sample trees were divided into three biomass components: (1) stem; (2) large branches (>2 cm diameter); and (3) small branches (<2 cm diameter), along with foliage, buds, capsules, or flowers. These components were weighed using digital scales and fresh weight (kg) was recorded.
4.  For each tree, sub-samples (at least 2 kg) of these biomass components were taken to the laboratory for determining moisture content (MC%). From the base of bole to the height of the first limiting defect of each tree, a 40 mm wide disk was taken every 3.0 m for laboratory analysis.
5.  In the laboratory, the large branch and small branch samples were cut into small pieces and dried at 65–105 °C (as appropriate for the type of sample) until a constant weight was achieved. Stem disks were used to estimate green wood density ($\rho$, kg m$^3$) prior to drying.

The following additional steps were undertaken in the second dataset of 40 sample trees:

6.  Crown diameter (CD, m) was measured before felling the trees (at step 1). The CD measurements were taken for each tree using a tape measure, averaging the measurements from along and across the planting row.
7.  In the laboratory (at step 5), stem bark was removed from the disks, recording fresh weight of the bark and wood. The samples were dried, and oven-dry weight was determined. In addition, the average width of chainsaw cuts used to collect the discs was used to determine mass of sawdust based on the wood density ($\rho$ kg m$^{-3}$). The sawdust weight was added to the stem biomass. The formula for estimating stem bark and sawdust was described by Huynh et al. (2021) [45].

### 2.3. Data Analysis

#### 2.3.1. Variable Selection and Data Preparation

Identification of potential biomass models involves developing the relationships between aboveground biomass (AGB) and a combination of predictor variables. The response variable (AGB), dry weight of biomass (kg tree$^{-1}$) of each tree component, such as stems, branches and foliage were described by Huynh et al. (2021) [45]. Predictor variables were the respective diameter at breast height (D), height (H), wood density ($\rho$), crown diameter (CD) and crown volume (CV). The CV was calculated based on crown diameter (CD, m). We presumed that the spotted gum crowns could have many different solid shapes such as a sphere, ellipsoid, cylinder, cone, and paraboloid. These shapes were calculated using different formulas in the literature as outlined by Zhu et al. (2021) [46]. We tested five formulas based on five shapes, with the sphere being selected as the most appropriate and representative shape for spotted gum.

$$\text{CV}_{\text{sphere}} = \frac{4}{3}\pi\text{CD}^3 \tag{1}$$

where: CV is Crown volume (m$^3$) and CD is crown diameter (m).

#### 2.3.2. Model Fitting

A power-law equation $Y = \alpha X^{\beta} + \varepsilon$ was used to develop the allometric relationship between AGB (Y) and predictor variables (X). Power-law equations can be fitted as logarithmic transformations of the original data ln (Y) = ln(a) + ln(X), or as nonlinear models [47–49]. The application of logarithmic transformations is widely used for estimating

tree biomass [3,20,39,50,51]. However, testing general AGB models for tree biomass across Australia by Paul et al. (2016) [20] recommended that weighted nonlinear modeling should be used for tree diameters over 10 cm. Huynh et al. (2021) [44] compared log-linear and nonlinear equations for predicting belowground biomass (BGB) and results showed that the overall performance of weighted nonlinear models was better than log-linear models. This result also consists of findings from Huynh et al. (2019) [49], who reported that nonlinear models produce higher reliability. In addition, we also pre-analyzed AGB equations using Furnival's Index (FI) [50–52]. Preliminary results (Table S1) showed that FI values of weighted nonlinear models were lower than log-linear models. Hence in this study, we focused on weighted nonlinear regression models to develop AGB models. Weighted nonlinear models had the following general form:

$$AGB = \alpha \times X_{ij}^{\beta} + \varepsilon_{ij} \tag{2}$$

where AGB = total aboveground biomass kg tree$^{-1}$; $\alpha$ and $\beta$ are the parameters of the model; $X_{ij}$ is the covariate: D (cm), H (m), $\rho$ (kg m$^{-3}$) CD (m) and CV (m$^3$), or a combination of these variables for i$^{th}$ sampled tree; and $\varepsilon_{ij}$ is the random error related to the i$^{th}$ sampled tree. The variance function ($\varepsilon_{ij}$) [53,54] was described in Huynh et al. (2021) [44]. In this study, the weighting variables include D, D$^2$H, $\rho$, D$^2$HCD and D$^2$HCV.

Before adding height, wood density, crown diameter and crown volume to the diameter at breast height variate for AGB estimates, the models using five single variables (AGB = $\alpha \times$ D$^\beta$; AGB = $\alpha \times$ H$^\beta$; AGB = $\alpha \times$ CD$^\beta$ and AGB = $\alpha \times$ CV$^\beta$) were developed.

(a) Testing compound predictor variables including height and wood density

To determine the importance of different predictor variables and test the influence of height and wood density, we tested seven commonly used formulations (Table 2) with a dataset of 52 individual trees (dataset 1) including young trees and mature trees (D ranges from 11.8–42.0 cm) as (i) the combination of D and H, (ii) the inclusion of D and $\rho$ and (iii) combination of D, H and $\rho$ (Table 2).

**Table 2.** Type of predictor models used to develop biomass allometric equations: D is diameter at breast height (cm), H is total tree height (m), $\rho$ is wood density (kg m$^{-3}$), CD is crown diameter (m) CV is crown volume (cm$^3$), and $\delta$ is the variance function coefficient.

| Input Variable | Equation No. | Model Form | Weight Variable |
|---|---|---|---|
| Model set 1: Compound predictor variables including D, H and $\rho$, n = 52 trees | | | |
| D | (3) | AGB = $\alpha \times$ D$^\beta$ | $1/D^\delta$ |
| H | (4) | AGB = $\alpha \times$ H$^\beta$ | $1/H^\delta$ |
| D and H | (5) | AGB = $\alpha \times$ D$^\beta \times$ H$^{\beta 1}$ | $1/D^\delta$ |
| | (6) | AGB = $\alpha \times$ (D$^2$H)$^\beta$ | $1/(D^2H)^\delta$ |
| D and $\rho$ | (7) | AGB = $\alpha \times$ D$^\beta \times \rho^{\beta 1}$ | $1/D^\delta$ |
| D, H and $\rho$ | (8) | AGB = $\alpha \times$ D$^\beta \times$ H$^{\beta 1} \times \rho^{\beta 2}$ | $1/(D)^\delta$ |
| | (9) | AGB = $\alpha \times$ (D$^2$H$\rho$)$^\beta$ | $1/(D^2H\rho)^\delta$ |
| Model set 2a: Compound predictor variables including D, H, $\rho$ and CD, n = 40 trees | | | |
| D | (10) | AGB = $\alpha \times$ D$^\beta$ | $1/D^\delta$ |
| H | (11) | AGB = $\alpha \times$ H$^\beta$ | $1/H^\delta$ |
| CD | (12) | AGB = $\alpha \times$ CD$^\beta$ | $1/CD^\delta$ |
| D and CD | (13) | AGB = $\alpha \times$ D$^\beta \times$ CD$^{\beta 1}$ | $1/D^\delta$ |
| D, H and CD | (14) | AGB = $\alpha \times$ D$^\beta \times$ H$^{\beta 1} \times$ CD$^{\beta 2}$ | $1/D^\delta$ |
| | (15) | AGB = $\alpha \times$ (D$^2$HCD)$^\beta$ | $1/(D^2HCD)^\delta$ |
| D, H, $\rho$ and CD | (16) | AGB = $\alpha \times$ D$^\beta \times$ H$^{\beta 1} \times \rho^{\beta 2} \times$ CD$^{\beta 3}$ | $1/D^\delta$ |
| | (17) | AGB = $\alpha \times$ (D$^2$H$\rho$ CD)$^\beta$ | $1/(D^2H\rho CD)^\delta$ |

**Table 2.** *Cont.*

| Input Variable | Equation No. | Model Form | Weight Variable |
|---|---|---|---|
| Model set 2b: Compound predictor variables including D, H, $\rho$ and CV, n = 40 trees | | | |
| CV | (18) | AGB = $\alpha \times CV^{\beta}$ | $1/CV^{\delta}$ |
| D and CV | (19) | AGB = $\alpha \times D^{\beta} \times CV^{\beta 1}$ | $1/D^{\delta}$ |
| D, H and CV | (20) | AGB = $\alpha \times D^{\beta} \times H^{\beta 1} \times CV^{\beta 2}$ | $1/D^{\delta}$ |
|  | (21) | AGB = $\alpha \times (D^2 HCV)^{\beta}$ | $1/(D^2 H\,CV)^{\delta}$ |
| D, H, $\rho$ and CV | (22) | AGB = $\alpha \times D^{\beta} \times H^{\beta 1} \times \rho^{\beta 2} \times CV^{\beta 3}$ | $1/D^{\delta}$ |
|  | (23) | AGB = $\alpha \times (D^2 H\rho CV)^{\beta}$ | $1/(D^2 H\rho CV)^{\delta}$ |

(b) Testing combinations of predictor variables including height, wood density and crown variables.

Using the dataset of 40 trees sampled in 2020 (dataset 2a and 2b) where CD and CV data were available, we explored whether the accuracy of the AGB models could be improved by adding CD or CV as predictor variables. We tested the (i) combination of CD and D, (ii) combination of D, H, and CD, and (iii) inclusion D, H, $\rho$ and CD. To compare the influence of these variables on the models in the same dataset, a new equation based on D and H alone (Equations (10) and (11)) was also created. A similar process was applied to the CV variable (Table 2).

### 2.3.3. Model Assessment and Selection

Candidate models were selected based on a combination of five fit statistics: (i) Akaike Information Criterion (AIC); (ii) adjusted $R^2$ (adj. $R^2$); (iii) average bias, (iv) root mean square error (RMSE) and (v) mean absolute percentage error (MAPE). In addition, diagnostic plots were also used to check for possible outliers and assess the goodness of fit of the models. The optimal models will have the lowest AIC, bias, RMSE and MAPE; low levels of collinearity as well as a high adj. $R^2$. These criteria were described by Huynh et al. (2021) [44].

The candidate nonlinear models were fitted by the weighted maximum likelihood procedure [55,56] using 'nlme' package in R and the plot diagnostics were checked using the ggplot 2 package in R [57].

### 2.3.4. Model Cross Validation

To assess the accuracy of selected biomass models against independent data, a cross validation procedure was undertaken by applying Monte Carlo cross validation (MCCV) [30,42,58]. Among the dataset 1, (Equations (3)–(9)) the MCCV was randomly split with 80% for training and 20% for testing [32] with the procedure repeated 100 times.

The statistics for validation of each model were averaged over the 100 realizations. For a smaller sample size of datasets 2a and 2b (Equations (10)–(23)), the MCCV procedure was repeated 40 times. The statistics for validation of each model were averaged over the 40 realizations. Comparison models were also based on the same criteria in Section 2.3.3, including AIC, adj. $R^2$, percent bias, RMSE and MAPE [54,59,60]. Finally, a model with the fewest errors was selected as follows:

$$\text{Bias} = \frac{1}{R} \sum_{r=1}^{R} \frac{100}{n} \sum_{i=1}^{n} \frac{yi - \hat{y}}{yi} \tag{24}$$

$$\text{RMSE} = \frac{1}{R} \sqrt{\sum_{r=1}^{R} \sum_{i=1}^{n} (yi - \hat{y})^2} \tag{25}$$

$$\text{MAPE} = \frac{1}{R} \sum_{r=1}^{R} \frac{100}{n} \sum_{i=1}^{n} \frac{|yi - \hat{y}i|}{yi} \tag{26}$$

where R = number of resamplings; yi is observed AGB; and ŷ is estimated AGB from the cross-validation study.

To further test the application of our selected equation we used an independent AGB dataset for 86 individual trees [61] of CCV and *Corymbia maculata* (two different species of spotted gum) collected from across Australia. The validation procedure was also repeated 100 times and the above criteria were used to assess the model.

## 3. Results

### 3.1. Basic Measurements and Tree Component Biomass

The tree over bark diameters (D) ranged from 11.8 cm to 42.0 cm across all sites and ages, while the height (H) varied from 12.6 m to 32.0 m (Table 1). The minimum crown diameter (CD) was 2.8 m and the maximum was 9.9 m. The mean basic wood density ($\rho$) of the sapwood and heartwood without bark was 663.8 kg m$^{-3}$ at age of nine in trial 451D and this value increased to 736.5 kg m$^{-3}$ by age 20 at the same site (Table 1). There was significant variability in wood density among sites and ages ($p$-value = 0.007; df = 4; F-value = 3.95). The individual tree AGB among the three sites ranged from 43.9 to 1503.7 kg tree$^{-1}$ (Table 3).

**Table 3.** Biomass of each tree component, including stem (under bark), stem bark, large branches ($\geq$2 cm diameter), small branches (<2 cm diameter) and leaves sampled at three sites (451D, 451G, and 13PHY). For each component mean, minimum and maximum values are presented.

| Sites | n | Mean (min, max), kg | | | | |
|---|---|---|---|---|---|---|
| | | Stem | Bark | Large Branches | Small Branches and Leaves | Total AGB |
| 451G (7) | 3 | 67.8 (29.0–110.0) | 16.4 (8.7–23.4) | 10.6 (5.5–16.3) | 5.3 (2.3–7.2) | 100.0 (45.5–156.8) |
| 13PHY (8) | 6 | 70.8 (40.1–101.9) | 12.0 (7.9–16.1) | 28.5 (8.8–44.7) | 8.9 (4.6–13.4) | 120.2 (73.3–174.1) |
| 451D (9) | 3 | 59.4 (26.4–98.0) | 17.0 (10.8–24.6) | 3.6 (2.1–5.4) | 4.8 (3.3–7.7) | 84.8 (43.9–135.6) |
| 451G (18) | 13 | 417.1 (99.0–845.9) | 55.1 (19.8–109.9) | 179.1 (21.0–666.4) | 51.1 (9.6–109.7) | 702.5 (149.4–1503.7) |
| 451D (20) | 27 | 329.9 (92.2–682.1) | 44.3 (18.8–75.5) | 159.1 (17.2–501.2) | 43.1 (7.2–172.5) | 576 (149.7–1431.3) |
| Total | 52 | 291.1 (26.4–845.9) | 40.1 (7.9–109.9) | 131.5 (2.1–666.4) | 36.8 (2.3–172.5) | 499.4 (43.9–1503.7) |

The weight of AGB varied by different ages and sites; minimum AGB of young trees ranged from 43.9 to 156.8 kg tree$^{-1}$, whereas maximum AGB of mature trees ranged from 149.4 to 1503.7 kg tree$^{-1}$. There was variation among tree components, the weight of stem was higher than other components such as large branches, small branches, and leaves (Table 3). The relative proportions of the aboveground tree biomass components are presented in Figure 1. The highest proportion of biomass was in potentially commercial logs (60.3%) while the smallest proportion was small branches and leaves.

### 3.2. Data Exploration and Variable Selection

Before fitting the allometric equations, a correlation matrix plot was visualized to check assumptions and explore the strength of relationships between response variable (AGB) and predicted variables (D, H, CD, $\rho$ and CV) (Figure 2). These relationships were tested based on the AGB and natural logarithm of five variables (ln(D), ln(H), ln(CD), ln($\rho$) and ln(CV)) using Spearman default method. These predicted variables displayed positive correlations (note there were no negative correlations, hence only blue circles displayed in this figure). The D, H, CD and CD were strongly correlated with AGB, while $\rho$ by itself had a weaker correlation (color intensity and the size of the circle are less) with AGB. However, these relationships were tested based on every single variable, the nonlinear relationships and fitted statistics for the equations were observed after applying weighted nonlinear methods.

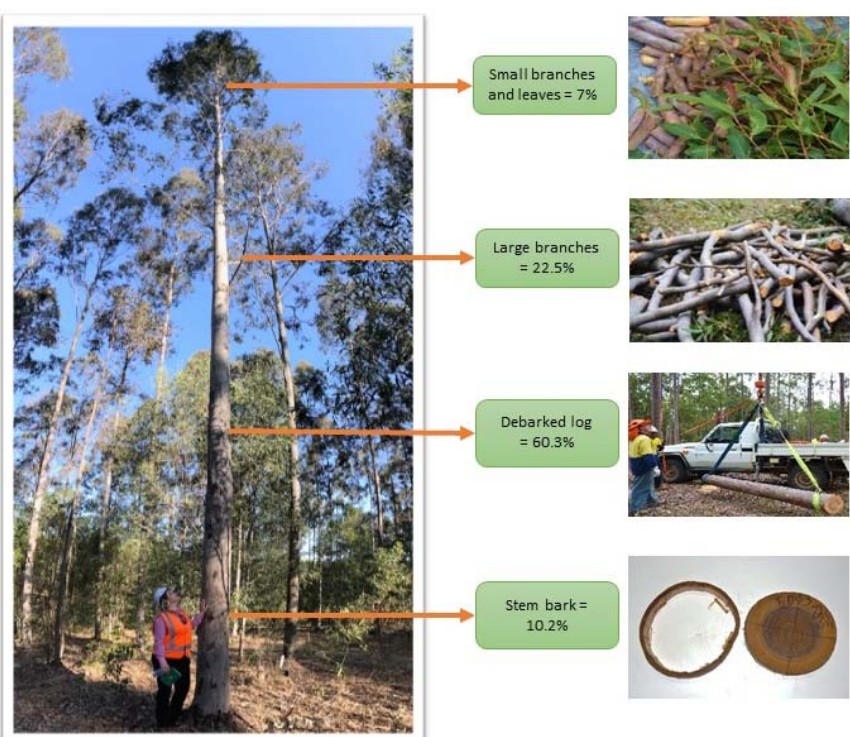

**Figure 1.** Average proportions of aboveground biomass components of plantation grown spotted gum (*Corymbia citriodora* subsp. *variegata*) trees, based on data from 52 destructively sampled trees. Please note that debark log biomass was determined from the weight of logs with their bark and the proportions of stem bark and wood estimated indirectly from stem discs from these logs [45].

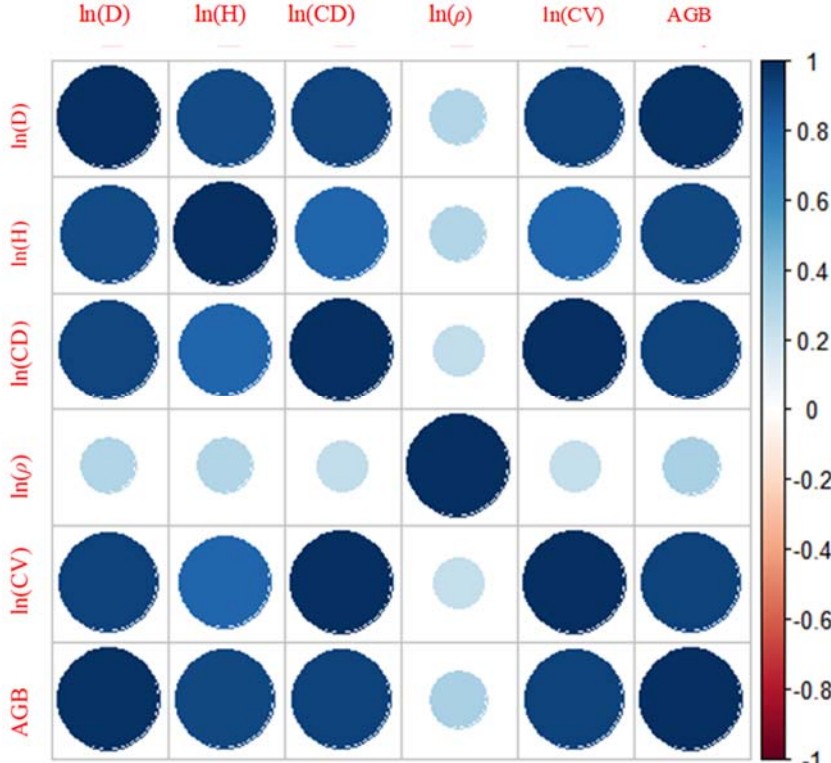

**Figure 2.** Positive correlation between response variable AGB and natural logarithm of five predicted variables D, H, CD, ρ and CV with Spearman rank correlation from −1 to +1. The color intensity and the size of the circle are proportional to the correlation coefficients.

### 3.3. Allometric Equations for AGB

The preliminary analysis illustrated that logarithmically transformed models had higher FI values compared to weighted nonlinear models (Table S1). The FI of log-linear models ranged from 36.7 to 128.7 whereas weighted nonlinear models gave very small values of FI (0.0004–0.0316), (Table S1).

The main statistics used to assess reliability of AGB models are presented in Table 4, comparative plots of predicted and observed data of Equations (3)–(9) are presented in Figure 3. These equations accounted for 61–98% of the variation in AGB, with AICs ranging from 442 to 670. Bias, RMSE and MAPE were similar for models with D, H, and $\rho$ (Table 4). The results showed that predictor D alone (Equation (3)) performed strongly across all predicted models based on AIC (541.1) whereas H had the poorest relationship with AGB. However, bias was quite small in the model with H alone. Both these equations appear to be reasonably reliable and are investigated further along with the other equations in the section on cross validation of results.

**Table 4.** Parameter estimates and their standard errors for AGB models developed based on weighted nonlinear models: D is diameter at breast height (cm), H is total tree height (m), AIC is Akaike Information Criterion, bias is averaged bias (%), RMSE is averaged root mean square error (kg), MAPE is averaged mean absolute percent error (%). Each equation number here refers to those in Table 2.

| Equation No. | Parameter Estimates | | | | | AIC | Adj. $R^2$ | Bias (%) | RMSE (kg) | MAPE (%) |
|---|---|---|---|---|---|---|---|---|---|---|
| | $\alpha$ | $\beta$ | $\beta_1$ | $\beta_2$ | $\beta_3$ | | | | | |
| Model set 1: Compound predictor variables including D, H and $\rho$ (n = 52 trees) | | | | | | | | | | |
| (3) | 0.08220 | 2.64134 | | | | 544.1 | 0.963 | −0.0025 | 0.0200 | 0.0085 |
| (4) | 0.00622 | 3.49873 | | | | 670.8 | 0.720 | 0.0001 | 0.0034 | 0.0012 |
| (5) | 0.05251 | 2.40238 | 0.38285 | | | 546.3 | 0.973 | −0.0023 | 0.0316 | 0.0132 |
| (6) | 0.02533 | 1.00656 | | | | 554.0 | 0.975 | 0.0212 | 0.0500 | 0.0186 |
| (7) | 0.05252 | 2.40266 | 0.38253 | | | 546.3 | 0.973 | −0.0023 | 0.0316 | 0.0132 |
| (8) | 0.00233 | 2.42585 | 0.30576 | 0.49890 | | 551.8 | 0.972 | 0.0001 | 0.0248 | 0.0106 |
| (9) | 0.00004 | 0.99037 | | | | 561.6 | 0.963 | 0.0004 | 0.0004 | 0.0002 |
| Model set 2a: Compound predictor variables including D, H, $\rho$ and CD (n = 40 trees) | | | | | | | | | | |
| (10) | 0.10606 | 2.56803 | | | | 442.3 | 0.950 | 0.0000 | 0.0043 | 0.0009 |
| (11) | 0.00027 | 4.45063 | | | | 545.9 | 0.614 | 0.0000 | 0.0002 | 0.0000 |
| (12) | 33.24309 | 1.61825 | | | | 532.6 | 0.769 | 4.3446 | 30.0835 | 6.1784 |
| (13) | 2.30247 | 1.07425 | | | | 450.2 | 0.947 | −0.0003 | 0.0032 | 0.0007 |
| (14) | 0.05153 | 2.18627 | 0.54648 | 0.11719 | | 456.7 | 0.964 | 0.0007 | 0.0259 | 0.0050 |
| (15) | 0.19568 | 0.68009 | | | | 460.3 | 0.961 | −1.0183 | 1.6823 | 0.3358 |
| (16) | 0.00079 | 2.07194 | 0.69202 | 0.60292 | 0.18009 | 463.1 | 0.967 | 0.0031 | 0.0318 | 0.0060 |
| (17) | 0.00156 | 0.69886 | | | | 455.2 | 0.965 | 0.0347 | 0.1618 | 0.0326 |
| Model set 2b: Compound predictor variables including D, H, $\rho$ and CV (n = 40 trees) | | | | | | | | | | |
| (18) | 15.35139 | 0.53941 | | | | 532.6 | 0.781 | 2.7223 | 18.9797 | 3.8982 |
| (19) | 0.09881 | 2.61161 | −0.01109 | | | 450.2 | 0.950 | −0.0003 | 0.0032 | 0.0007 |
| (20) | 0.04872 | 2.18625 | 0.54650 | 0.03907 | | 456.7 | 0.966 | 0.0007 | 0.0259 | 0.0050 |
| (21) | 0.95382 | 0.38323 | | | | 496.5 | 0.908 | −0.5644 | 5.6982 | 1.1344 |
| (22) | 0.00072 | 2.07191 | 0.69205 | 0.60293 | 0.06004 | 463.1 | 0.970 | 0.0030 | 0.0318 | 0.0060 |
| (23) | 0.06581 | 0.38942 | | | | 494.9 | 0.907 | 0.1717 | 1.2739 | 0.2542 |

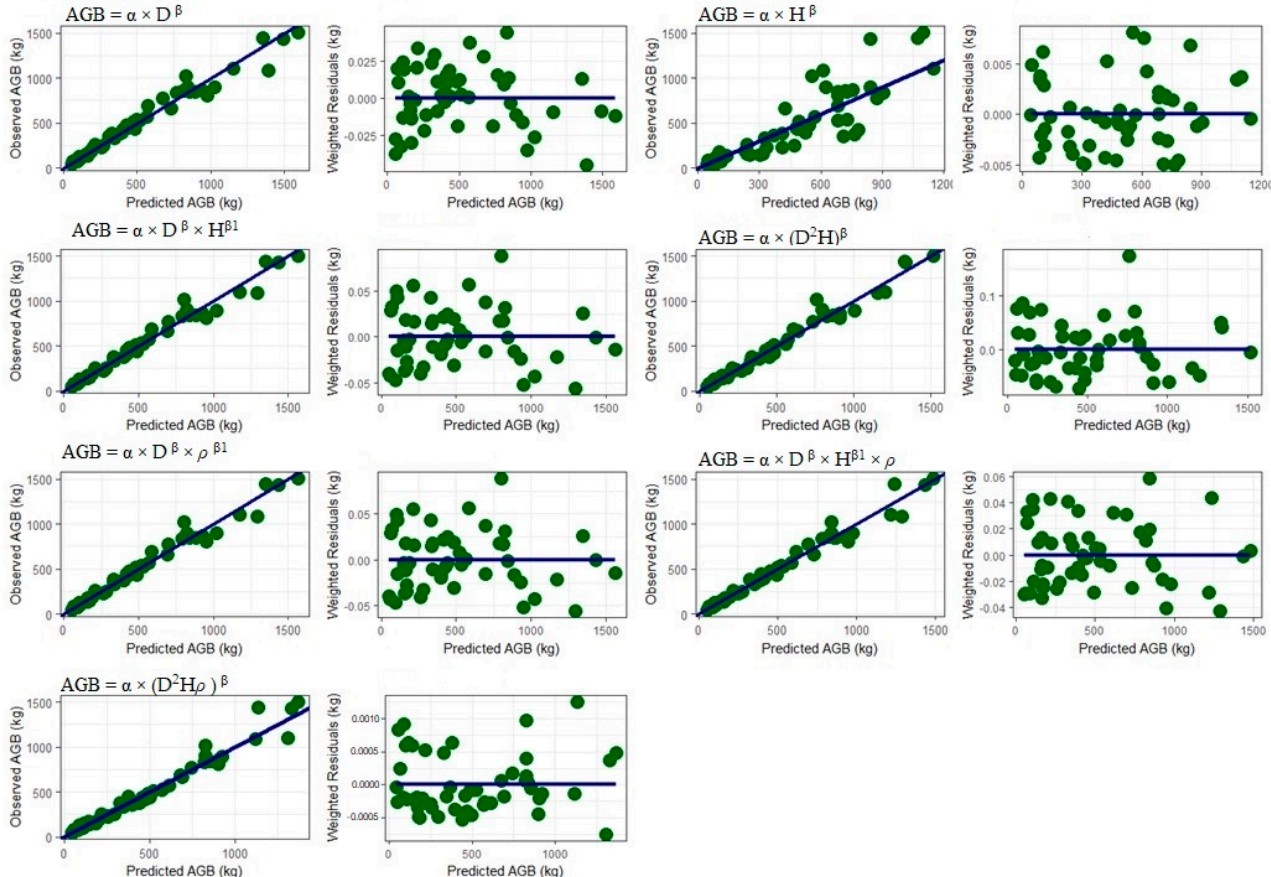

**Figure 3.** Relationship between aboveground biomass (AGB, kg tree$^{-1}$ and predictor variables, D (cm), H (m) and $\rho$ (kg m$^{-3}$) and weighted residuals, for the candidate Equations (3) to (9) based on data for 52 sample trees. See Table 4 for criteria associated with these regressions.

### 3.3.1. Including Predictor Variables Height and Wood Density

Including the variables H and $\rho$ resulted in slight improvements (Table 4) relative to model with D alone (Equation (3)). The compound variate of D$^2$H (Equation (6)) improved adj. R$^2$ (0.975) in comparison to Equation (3) with an adj. R$^2$ = 0.963. The combination of D$^\beta$ × H$^{\beta 1}$ (Equation (5)) and D$^\beta$ × $\rho^{\beta 1}$ (Equation (7)) provided similar goodness of fit criteria (AIC = 546.3, adj. R$^2$ = 0.973 and MAPE = 0.0132 %). Adding three predictor variables D, H and $\rho$ into the model (Equations (8) and (9)) reduced RMSE and MAPE, but it did not improve the AIC compared with Equation (3).

### 3.3.2. Including CD and CV in Biomass Equations

Model statistics in dataset 2a are presented in Table 4 and diagnostic plots are presented in Figure S2a,b. Addition of CD as a compound predictor variable in Equations (12)–(17) did not improve model performance compared with the model of D alone (Equation (10)), except for adj. R$^2$. The adj. R$^2$ increased when adding a combination of CD-H and CD-H-$\rho$, with the changes in adj. R$^2$ from 0.950 (Equation (12)) to 0.967 (Equation (16)).

Starting with the six base Equations (12)–(17) above, we then added terms to test for CV (Equations (18)–(23)) in dataset 2b (Table 3). The model based on only CV (Equation (18)) gave the poorest fit as indicated by the large AIC (532.6), while the combination D × CV (Equation 19) had the smallest AIC (450.2). However, use of multiple variables, including D-H-CV-$\rho$ resulted in a superior adj. R$^2$ (0.970) relative to the combination of D and CV. Using CD (Equation (12)) and CV alone (Equation (18)) had the same value of AIC (532.6), but Equation (18) improved AGB estimates, increasing adj. R$^2$ and reducing bias, RMSE and MAPE.

### 3.4. Cross Validation Biomass Models

The Monte Carlo cross validation (MCCV) procedure was used to evaluate and improve predictive performance of Equations (3)–(23). Validation results of these equations are provided in Table S2 and diagnostic plots are present in Figure 4. Cross validation results of models using the four single predictor variables D, H, CD and CV are presented in Table 5.

#### 3.4.1. Models Using Diameter, Height and Wood Density

With dataset 1, the accuracy of AGB prediction between Equations (3)–(9) varied significantly. The model created using H alone (Equation (4)) had relatively high errors (averaged AIC = 533.1, bias = −41.7%, MAPE = 55.3% and adj. $R^2$ = 0.642) while the model using D (Equation (3)) proved to be the best (Table 5), with the lowest values of AIC = 434.4, bias = −2.2% and MAPE = 7.2% (Table 5). Adding H and $\rho$ as a second or third variable improved the adj. $R^2$ (Δ adj. $R^2$ from 0.099 to 0.135) whereas other criteria such as AIC, bias and MAPE did not improve (Table 5).

#### 3.4.2. Models Using Crown Diameter and Crown Volume

For validated models using CD in the second dataset (Equations (12)–(17)), results showed that using CD markedly increased errors for predicting AGB compared with the D-based model (Equation (10)), with the highest ΔAIC = −73.2 and ΔMAPE = −18.6% (Equation (12)) (Table 5). Similarly, all models with CV (Equations (18)–(23)) performed poorly in comparison to Equation (10), with ΔAIC = −33.1 and ΔMAPE = −7.2%. Using CV alone in Equation (18) estimated AGB slightly better than the CD in Equation (12), with AIC = 2.1 smaller and MAPE lowered by 2.3%. By contrast, models with CV added as a second variable showed less accuracy than adding CD. For example, the MAPE ranged from 7.1–13.3% when applying CD, while ranges of 10.7–17.1% were seen for the CV models (Table 5).

#### 3.4.3. Cross Validation against an Independent Dataset

Using our selected equation based on D (Equation (3)) with the dataset of Paul et al. (2016) [61] AGB resulted in bias = −18.0%, RMSE = 0.3 kg and MAPE = 25.8%. Even though the sample trees were collected from two species planted across a wide range of environments and ecoregions (including temperate broadleaf and mixed forests, Mediterranean forests, woodlands, and scrub woodlands), these results provide greater confidence in Equation (3) to predict biomass for these species grown in Australia.

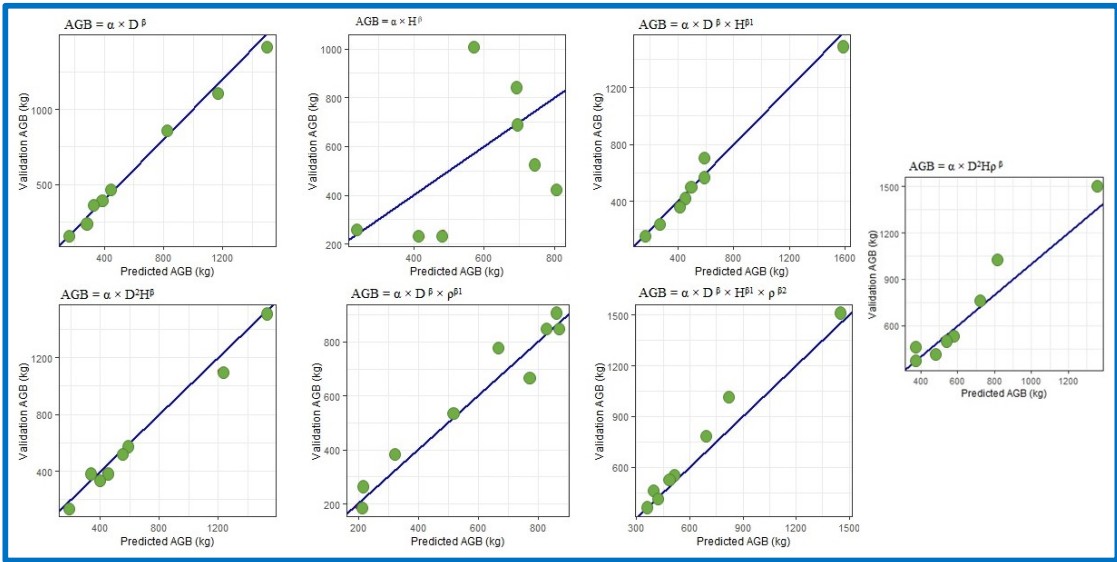

**Figure 4.** *Cont.*

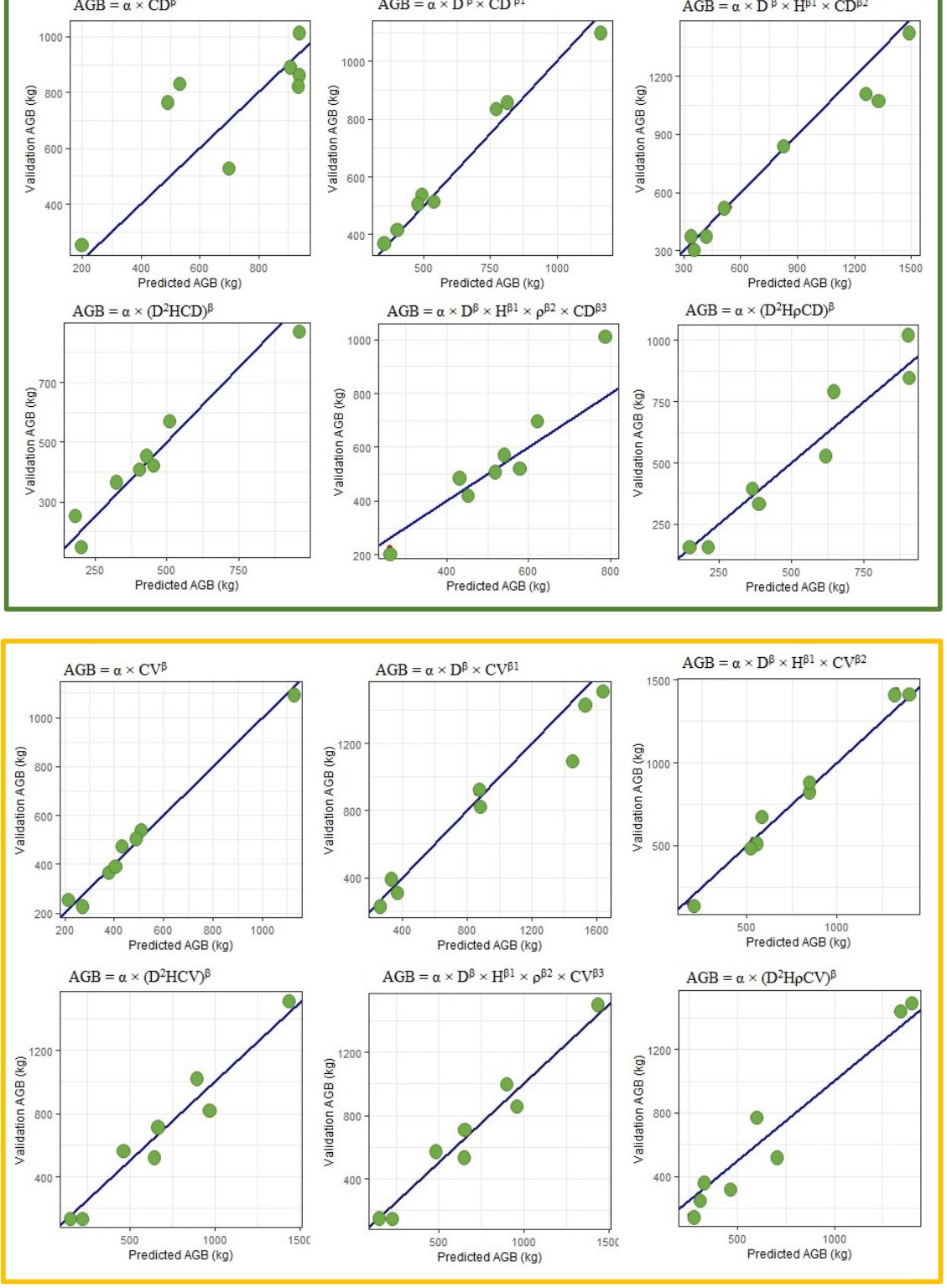

**Figure 4.** Observed and predicted plots for AGB models validated for all biomass models using MCCV method. Plots with blue border were validated for Equations (3)–(9) with compound predictor variables of D-H-$\rho$, cross validation procedure was used 80% data used for training, 20% data for testing, the process is repeated 100 times; Plots with green border validated and repeated 40 times for Equations (12)–(17) with compound predictor variables D-H-$\rho$-CD. The same process was applied for Equations (18)–(23) with compound predictor variables D-H-$\rho$-CV and plots with orange border. See Table S2 for criteria associated with these models.

**Table 5.** The comparison of cross validation results when the model based on D alone (Equations (3) and (10)) was compared with models using different predictor variables. Equation (3) was compared with Equations (4)–(9), and (10) was compared with Equations (11)–(23). A negative change in AIC, RMSE and MAPE indicates that the D-based equation is better than others. A positive change in adj. $R^2$ means the model D-based is superior. We also present the five equations here (Equations (3), (4), (10), (12) and (18)) to estimate biomass directly from D, H, CD and CV.

| Equation No. | Model Form | AIC | Adj. $R^2$ | Bias | RMSE | MAPE |
|---|---|---|---|---|---|---|
| (3) | $AGB = \alpha \times D^\beta$ | 434.4 | 0.823 | −2.2 | 0.115 | 7.2 |
| (4) | $AGB = \alpha \times H^\beta$ | 533.1 | 0.642 | −41.7 | 0.679 | 55.3 |
| (10) | $AGB = \alpha \times D^\beta$ | 357.2 | 0.880 | −6.0 | 0.114 | 6.8 |
| (12) | $AGB = \alpha \times CD^\beta$ | 430.4 | 0.964 | −6.5 | 0.428 | 25.4 |
| (18) | $AGB = \alpha \times CV^\beta$ | 428.3 | 0.964 | −14.8 | 0.348 | 23.1 |
| | | Δ AIC | Δ Adj. $R^2$ | Δ Bias | Δ RMSE | Δ MAPE |
| Model set 1: Compound predictor variables including D, H and $\rho$ | | | | | | |
| (4) | $AGB = \alpha \times H^\beta$ | −98.7 | 0.181 | 39.4 | −0.564 | −48.1 |
| (5) | $AGB = \alpha \times D^\beta \times H^{\beta 1}$ | −6.7 | −0.099 | 1.2 | 0.016 | −0.1 |
| (6) | $AGB = \alpha \times (D^2 H)^\beta$ | −13.2 | −0.149 | 6.2 | −0.011 | −3.4 |
| (7) | $AGB = \alpha \times D^\beta \times \rho^{\beta 1}$ | −4.0 | −0.098 | 1.1 | 0.011 | 0.8 |
| (8) | $AGB = \alpha \times D^\beta \times H^{\beta 1} \times \rho^{\beta 2}$ | −13.4 | −0.130 | 1.9 | 0.021 | 0.7 |
| (9) | $AGB = \alpha \times (D^2 H \rho)^\beta$ | −19.8 | −0.135 | 8.6 | −0.023 | −4.2 |
| Model set 2a: Compound predictor variables including D, H, $\rho$ and CD | | | | | | |
| (11) | $AGB = \alpha \times H^\beta$ | −86.9 | 0.255 | 9.9 | −0.096 | −11.4 |
| (12) | $AGB = \alpha \times CD^\beta$ | −73.2 | −0.084 | 0.5 | −0.315 | −18.6 |
| (13) | $AGB = \alpha \times D^\beta \times CD^{\beta 1}$ | −7.8 | 0.054 | 0.2 | −0.012 | −0.3 |
| (14) | $AGB = \alpha \times D^\beta \times H^{\beta 1} \times CD^{\beta 2}$ | −19.7 | 0.003 | −1.2 | 0.021 | −0.7 |
| (15) | $AGB = \alpha \times (D^2 HCD)^\beta$ | −16.6 | −0.084 | −2.6 | −0.010 | −0.9 |
| (16) | $AGB = \alpha \times D^\beta \times H^{\beta 1} \times \rho^{\beta 2} \times CD^{\beta 3}$ | −24.4 | −0.047 | −1.7 | 0.017 | −1.7 |
| (17) | $AGB = \alpha \times (D^2 H \rho CD)^\beta$ | −12.8 | −0.083 | −2.9 | −0.075 | −6.5 |
| Model set 2b: Compound predictor variables including D, H, $\rho$ and CV | | | | | | |
| (18) | $AGB = \alpha \times CV^\beta$ | −71.1 | −0.084 | 8.8 | −0.234 | −16.3 |
| (19) | $AGB = \alpha \times D^\beta \times CV^{\beta 1}$ | −9.3 | −0.046 | 0.2 | −0.026 | −4.2 |
| (20) | $AGB = \alpha \times D^\beta \times H^{\beta 1} \times CV^{\beta 2}$ | −16.2 | 0.004 | −1.2 | −0.031 | −4.5 |
| (21) | $AGB = \alpha \times (D^2 HCV)^\beta$ | −45.7 | −0.084 | −2.2 | −0.039 | −4.2 |
| (22) | $AGB = \alpha \times D^\beta \times H^{\beta 1} \times \rho^{\beta 2} \times CV^{\beta 3}$ | −11.8 | −0.047 | −1.7 | −0.026 | −3.9 |
| (23) | $AGB = \alpha \times (D^2 H \rho CV)^\beta$ | −44.2 | −0.084 | −2.4 | −0.039 | −10.3 |

## 4. Discussion

The development of carbon accounting models for Tier 2 and 3 methods can be complex as they require information on model accuracy and reliability [27]. There is an initial high cost associated with sampling to collect adequate biomass data to develop, evaluate and apply the methodology [7]. However, if allometric equations (required by Tier 2 and 3) can be developed, carbon accounting will have higher reliability and an overall reduction in costs [3,27,62]. Furthermore, the advances in technology such as remote sensing, combined with appropriate allometric equations based on variables that can be readily estimated (e.g., tree height, diameter, crown volume), should lead to improved predictions of plantation scale biomass [23,27].

### 4.1. Equation Development and Cross Validation

One of the most common mistakes identified in biomass determinations indicated by Sileshi (2014) [41] was the choice of analytical methods to develop allometric equations. Most studies (66% of all allometric equations) use log-linear models and authors often provide little evidence for choosing a particular method [41]. Our study found that weighted

nonlinear models had significantly smaller FI values than log-linear models (Table S1) across all models tested. This finding supports the conclusion of Huynh et al. (2021) [44] that weighted nonlinear equations had relatively low error values compared to log-linear equations. In some cases, the log-linear models may be suitable for estimating biomass of small-diameter trees (D < 10 cm), but these models may not always be suitable for estimating biomass of larger trees [20]. Huynh et al. (2021) [44], indicated that applying log-linear models results in poor predictions of belowground biomass of trees with a $40^+$ cm diameter. Therefore, we suggest tree biomass equation development considers nonlinear models for large trees.

The main purpose of model validation is to avoid bias in prediction and assess the performance of biomass models [31]. This study indicates that although most developed models had small errors, validated models had higher errors (Table 5). The H model (Equation (4)), for example, had 0.0012% MAPE in the development model, whereas a 55.3% MAPE was found in the validation results. The model using H alone resulted in poor estimates of AGB in comparison with D alone. The evidence from this study is consistent with previous studies which involved H, indicating that the addition of H did not improve model performance [18,63]. Validation of our selected equation using D alone (Equation (3)) against independent data from Paul et al. (2016) [61] in this study provides confidence in the use of this equation at other spotted gum plantations. This is important especially given the high cost associated with destructive sampling [43].

### 4.2. Inclusion of Height and Wood Density

The addition of H and $\rho$ resulted in models with improved adj. $R^2$ values while other model criteria did not improve (Table 5). Findings from van Niekerk et al. (2020) [27] also showed that adding H and $\rho$ into allometric equations, resulted in the mean square error becoming only slightly higher than the D-based model. Our finding is also consistent with Sileshi (2014) [41] who found a small improvement in AGB prediction when H and $\rho$ were included. The highest adj. $R^2$ ($\Delta$ adj. $R^2 = -0.149$) was found with $D^2H$ in combination (Equation (6)). However, the largest adj. $R^2$ is not the most commonly used criteria to determine the best model, as the addition of polynomial terms increases the adj. $R^2$ [20,41]. Some authors [64–69] found that $D^2H$ improved the accuracy of biomass equations. By contrast, other studies agreed that D-based allometry is a better predictor of AGB [20,27,41,70,71] based on a combination of criteria (AIC, adj. $R^2$, bias and MAPE).

It was found that D and $\rho$ in combination (Equation (7)) tended to be better (with on $\Delta$ AIC = $-4.0$, $\Delta$ bias = $\pm$ 1.1% and $\Delta$ MAPE = 0.8%) than a combination D and H (Equation (5)) (Table 5). Paul et al. (2016) [20] found that adding $\rho$ did not improve biomass models. It is important to note that the wood density data in Paul's study had some potential limitations: 88% of $\rho$ values were collected from the Global Wood Density Database with only 12% of the data being estimated based on field measurements. In our case, all $\rho$ values were estimated using multiple disks per sample tree, with a total of 223 disks from the 40 mature trees sampled in 2020 [45] and 12 sample trees in 2009. The slightly improved predictive ability of models with $\rho$ may be due to the $\rho$ values in our data being significantly different (*p*-value = 0.007) between age groups and sites. Wood density also varies among species and positions in the stem. Therefore, further study should investigate the influence of wood density for different species when developing biomass models.

### 4.3. Influence of Crown Diameter and Crown Volume

The fact that D is relatively easy to measure in forest inventory, and has low measurement error [3,41] has led many to consider it the most useful variable for biomass prediction [18,20,27,69]. However, with the advances in remote sensing, alternative variables (e.g., crown dimensions) could be readily included in biomass equations to estimate large-scale (e.g., plantation wide) biomass accumulation [1,2,69].

The use of LiDAR for crown volume estimation could become increasingly important as a predictor variable as this variable can be easily derived from LiDAR inventories [46]. However, only a few studies were conducted using this variable [23,27] and these studies have not validated the predictive performance of the model against new data [69]. Furthermore, there are more variables that could be derived from LiDAR data that could be considered to ensure greater accuracy of biomass values [14,20,23]. Our validation results showed that $\Delta$ AIC = $-86.9$ and $\Delta$ adj. $R^2$ = 0.255 in Equation (11) while $\Delta$ AIC = $-71.1$ and $\Delta$ adj. $R^2$ = $-0.084$ in the Equation (18), and diagnostic plot (Figure S3) indicating narrow variation (Table 5 and Table S3). This finding is consistent with the results of Goodman et al. (2014) [69] and van Niekerk et al. (2020) [27] who found that using crown radius of large trees improved predicted biomass models more than height. Validation models using our data imply that using crown volume was particularly important for building confidence in estimating biomass/carbon and could potentially be applied to accurately predict AGB using remote sensing tools such as LiDAR.

*4.4. Evaluating Existing Applicability Models*

We estimated the AGB of our sample trees using four published models which were also derived using destructive sampling methods for large eucalypt trees. Models were developed for (1) tropical and subtropical eucalypt woodlands [18]; (2) native spotted gum (*Corymbia maculata*) forests [22]; (3) hardwood trees from the genera of *Eucalyptus*, *Corymbiam* and *Angophora* [20]; and (4) *Eucalyptus grandis* × *E. nitens* hybrids [27].

The MCCV method was undertaken to validate the performance of these published equations for our dataset. The MCCV was run 100 times using 20% random splitting data. Averaged errors of these equations are presented in Table S3. A graph showing the comparisons between the AGB model in this study, the model of Ximenes et al. (2006) [22] and the model of Paul et al. (2016) [20] are presented in Figure 5. The smaller errors (bias, RMSE and MAPE) indicate the accuracy of the model based on D (Equation (3)) for predicting AGB. The averaged errors for individual trees of the previously published equations were always higher than that of Equation (3) (Table S3), with MAPE of the published models ranging from 18.8–30.9%. For a closely related spotted gum species (*Corymbia maculata*), the estimated AGB using the model of Ximenes et al. (2006) [22] resulted in under-estimation of AGB; with bias =18.6% and MAPE of 18.8%, while other models had higher errors than Ximenes's equation (Figure 5). This suggests that applying equations to data from closely related species in the same genus could reduce predicted AGB errors.

The datasets of Williams et al. (2005) [18] were mostly from trees of *Eucalyptus crebra* (18 trees), *E. foelscheana* (20 trees), and *E. terminalis* (20 trees). These species had lower heights than our sample trees despite having a similar median diameter and biomass values. For example, *E. crebra* with an average D = 26.3 cm, H = 17.1 m and AGB = 492.4 kg, compared to our trees with an average D = 25.9 cm, H = 23.9 m, AGB = 499.4 kg. Although William's data were collected across the Northern Territory, Queensland and New South Wales (Australia), AGB is underestimated when applying their equations to our data, with bias = $-30.9$%, RMSE = 0.2 and MAPE = 30.9%. This corroborates previous findings that allometric relationships for prediction of AGB may vary due to species characteristics [72], environment, forest structure [73] and wood density [74].

The validation results for the biomass model of van Niekerk et al. (2020) [27] developed in South Africa for *E. grandis* × *E. nitens* hybrids, suggest that applying this equation results in lower errors than the equation of Paul et al. (2016) [20], with bias = 21.8%, RMSE = 0.2 and MAPE = 21.8% (Table S3). A possible explanation for this result may be due to the differences between natural forests and plantations. Paul's model was developed based mostly on data collected from natural forests across Australia. The advantage of the Paul model is that it can be applied widely, but as a consequence of this, the errors are higher with such a model. Another difference may be due to the method used for developing the models. The equation of van Niekerk et al. (2020) [27] was developed using the weighted

nonlinear method, while the equation of Paul et al. (2016) [20] used the log-linear method. The issues associated with selection of biomass models are reviewed by Sileshi (2014) [41] who points out some of the pitfalls of using the log-linear methodology.

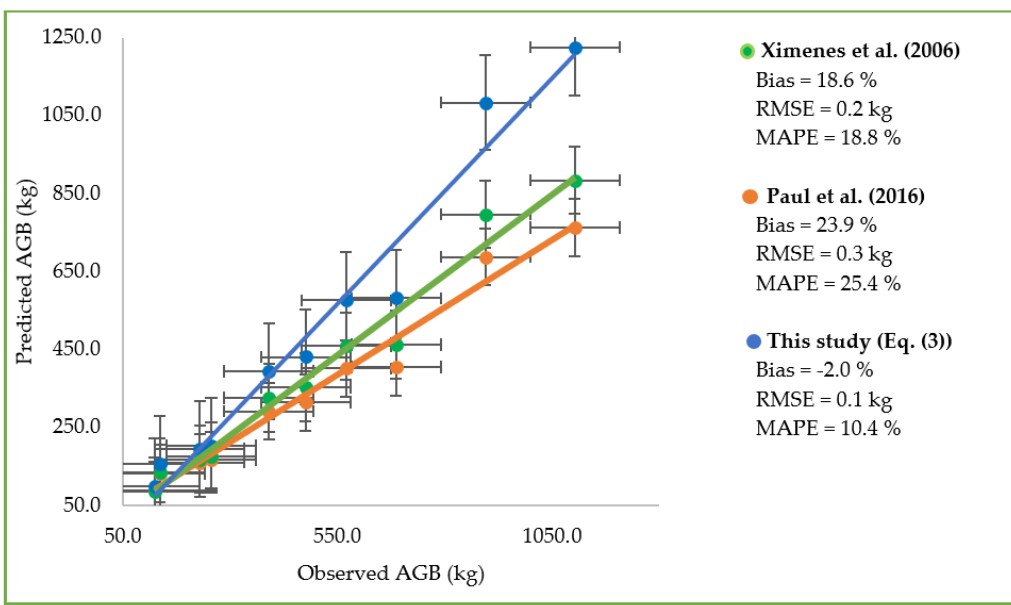

**Figure 5.** Validation and comparison of the D-based equation of this study and published equations in different regions in Australia, with spotted gum native forests [22] and mixed hardwood forests of Eucalyptus, Corymbia and Angophora [20]. The dataset was validated by the MCCV method with 20% random splitting data and the process was repeated 100 times to average the errors.

## 5. Conclusions

When developing and testing allometric equations, it will be necessary to estimate biomass using the higher tier methods of the IPCC. The models reported in this study followed high-tier methods and could be used to accurately predict the biomass and carbon sequestered in spotted gum (CCV) plantations relatively easily, with minimal costs and time, compared to destructively sampling trees. The equations developed here were independently validated, and we tested the application of these models more broadly with CCV and *Corymbia maculata* from natural forests in NSW.

This study showed that Equation (3), AGB = $\alpha \times D^{\beta}$, performed better than equations with other variables such as H, $\rho$, CD, and CV, with the lowest values of AIC, bias and MAPE. Adding more tree variables to the model led to increased adj. $R^2$ while other criteria did not improve when compared with the D-based model. A combination of D and $\rho$ in Equation (7) had slightly improved errors compared to the equation with D and H (Equation (5)). As an alternative to the D-based equation, an equation using CV (Equation (18)) was better than applying models based on H or CD. However, additional work is needed to test the application of crown variables to predict biomass in unrepresented regions and species.

While AGB can be predicted more accurately using D from simple forest measurements, measurement of D is not practical over large areas due to the high cost associated with extensive field data measurements. Remote sensing, using techniques such as radar or LiDAR, can provide data, such as that on crown diameter at various scales, from local to global scales. Applying the allometric equation based on crown volume in combination with remote sensing datasets would therefore allow broad-scale biomass estimation in spotted gum forests.

**Supplementary Materials:** The following are available online at https://www.mdpi.com/article/10.3390/f13030486/s1, Figure S1. Distribution of diameter at breast height (left) and total height

(right) of 52 sample trees used to develop biomass equations; Figure S2. (a) Plots of biomass estimation models based on data set 2a. With compound predictor variables of D, H, ρ and CD for 40 samples trees. See Table 4 for criteria associated with these regressions; (b) Plots of biomass estimation models based on data set 2b. With compound predictors variable of D, H, ρ and CV for 40 samples trees. See Table 4 for criteria associated with these regressions; Table S1. The Furnival's Index (FI) used to compare logarithmically transformed models and weighted nonlinear models. The lower FI indicated more reliable models; Table S2. Average predicted error of biomass equations using Monte Carlo cross-validation (MCCV), the procedure was used 80% data used for training, 20% data for testing, the process is repeated 100 times for Equations (3)–(9) and 40 times for Equations (10)–(23); Table S3. Comparison of average errors of Equation (3) in this study and published AGB models for eucalypt species.

**Author Contributions:** T.H.: Conceptualization, methodology, formal analysis and writing original draft. T.L.: Conceptualization, investigation, writing—review & editing, supervision. G.A.: Conceptualization, investigation, writing—review & editing, supervision. A.N.A.P.: Investigating, writing—review & editing. D.J.L.: Conceptualization, investigation, resources, writing—review & editing, supervision. All authors have read and agreed to the published version of the manuscript.

**Funding:** This research was supported by the joint The Ministry of Education and Training, Vietnam (MoET) and University of the Sunshine Coast (USC) scholarship (MoET-VIED/USC); Australian Government Research Training Program scholarship (RTP); and an internal grant provided by Forest Industries Research Centre (USC).

**Institutional Review Board Statement:** Not applicable.

**Informed Consent Statement:** Not applicable.

**Data Availability Statement:** Not applicable.

**Acknowledgments:** We thank HQPlantations for permission to destructively sample trees. We are especially grateful to the Department of Agriculture and Fisheries (DAF) for providing material and equipment for the fieldwork. We thank DAF staff, including Tracey Menzies, Tony Burridge and John Oostenbrink who contributed to field sampling and laboratory measurements. Mark Hunt (University of Tasmania) and Mila Bristow (Plant Health Australia), both formerly Queensland Government employees are thanked for contributing data from the destructive sampling of 7 to 9-year-old spotted gum trees. The authors thank Bao Huy, an Independent Consultant of Forest Resources and Environment-FREM for giving important suggestions on coding of equations. Helpful comments on the methodology from the Queensland Cyber Infrastructure Foundation are also acknowledged. We thank three reviewers for providing their useful suggestions and comments.

**Conflicts of Interest:** The authors declare that they have no known competing financial interests or personal relationships that could have appeared to influence the work reported in this paper.

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
