# Peer review of "Allometric Equations to Estimate Aboveground Biomass in Spotted Gum (Corymbia citriodora Subspecies variegata) Plantations in Queensland"

_forests, doi:10.3390/f13030486_

Round 1

Reviewer 1 Report

This work proposed a good way to estimating biomass and carbon accumulation for Corymbia citriodora subspecies variegata plantations in Queensland. The author suggested the key variables to estimating biomass. Some problems need the authors to revise:  

  1. The models parameters are based on the data from 52 sample trees which diameter at breast height are ranging from 11.8 to 42cm, then how about Allometric equations performance beyond this scope?
  2. Tree’s biomass should include the above and below ground biomass, then, how about the research progress on this part?
  3. The author got the conclusion that D-based equation have better performance than models based on other variables. Then, how can we predict aboveground biomass by combining remote sensing datasets?

Author Response

Dear reviewer,

We thank you for providing useful suggestions and comments to the manuscript. Your questions have been addressed as detailed below:

Q1. In our study, the allometric equation based on D in this study has been validated to against an independent dataset using Paul et al (2018) datasets (in section 3.4.3), with bias = -18.0%, RMSE = 0.3 kg and MAPE = 25.8%. In these datasets, D ranges from 3.8-52.2 cm and sample trees planted in a wide range of environments. In addition, we also validated the application of published equations of Ximenes et al. (2006) and Paul et al. (2016), William et al. (2005) and van Niekerk et al (2020) to our data. These previous equations were developed based on varied diameters (details of results presented on section of 4.4).  Based on the validation of our model to the previously published data and the validation of the published equations to the current data, it can be concluded that our equation performs well in ranges from small diameter to over 50 cm and this suit with tree size of plantation rotation. However, the application specific diameter for specific species in natural forests must be considered.

Q2. While we agree that biomass estimation should be included the above and belowground biomass. Simultaneously developing equations for both above and belowground components would be better. However, sampling for belowground biomass is time-consuming and high cost. We completed this work with data collection of 23 sample trees and it was published in Forests 12(9): 1210. The aboveground sampling was completed on the 52 trees included in this manuscript. Combining the two datasets to derive allometric equations would have resulted in a manuscript that was much longer (beyond the word limit of most journals). Hence, we developed equations for below and aboveground trees separately.

Q3. Even we concluded that equation of D-based performs better than other variables. However, we also developed and validated equations with a crown diameter and crown volume variables. Results showed that we might use a single variable of crown volume to estimate biomass using remote sensing datasets as some techniques such as radar or LiDAR could apply to collect data of crown trees.  

I hope these explanations above could clarify your concerns.

Cheers,

Trinh

Reviewer 2 Report

The manuscript is well prepared and interesting. The scientific soundness is on a high level.

I found just minor typo:

Line 136 (Table 1 title): instead of "crow" should be "crown".

Author Response

Dear reviewer,

We thank you for providing useful suggestions and comments to the manuscript. Your suggestions have been addressed as detailed below:

Line 136: The word "crown" was revised

Cheers,

Trinh

Reviewer 3 Report

Great job! Most of my remarks are of technical type and should be easy to implement (if you decide to do it, naturally). 

Best regards

Author Response

Dear reviewer,

We thank you for providing useful suggestions and comments to the manuscript. Your suggestions have been addressed. Please see the attached file below.

Cheers,

Trinh
